# Cannabinoids in Treating Chemotherapy-Induced Nausea and Vomiting, Cancer-Associated Pain, and Tumor Growth

**DOI:** 10.3390/ijms25010074

**Published:** 2023-12-20

**Authors:** Pavana P. Bathula, M. Bruce Maciver

**Affiliations:** School of Medicine, Stanford University, Palo Alto, CA 94305, USA; pavanapragnabathula@gmail.com

**Keywords:** cannabidiol, Δ-9-tetrahydrocannabinol, cancer, nausea, vomiting, pain management, antitumor

## Abstract

Cannabis has been used as an herbal remedy for thousands of years, and recent research indicates promising new uses in medicine. So far, some studies have shown cannabinoids to be safe in helping mitigate some cancer-associated complications, including chemotherapy-induced nausea and vomiting, cancer-associated pain, and tumor growth. Researchers have been particularly interested in the potential uses of cannabinoids in treating cancer due to their ability to regulate cancer-related cell cycle pathways, prompting many beneficial effects, such as tumor growth prevention, cell cycle obstruction, and cell death. Cannabinoids have been found to affect tumors of the brain, prostate, colon and rectum, breast, uterus, cervix, thyroid, skin, pancreas, and lymph. However, the full potential of cannabinoids is yet to be understood. This review discusses current knowledge on the promising applications of cannabinoids in treating three different side effects of cancer—chemotherapy-induced nausea and vomiting, cancer-associated pain, and tumor development. The findings suggest that cannabinoids can be used to address some side effects of cancer and to limit the growth of tumors, though a lack of supporting clinical trials presents a challenge for use on actual patients. An additional challenge will be examining whether any of the over one hundred naturally occurring cannabinoids or dozens of synthetic compounds also exhibit useful clinical properties. Currently, clinical trials are underway; however, no regulatory agencies have approved cannabinoid use for any cancer symptoms beyond antinausea.

## 1. Introduction

Marijuana, also known as hashish, ganja, bud, hemp, weed, or, more commonly, cannabis, is produced by female plants from three cannabinoid-containing species (indica, sativa, ruderalis) of the genus Cannabis. It is the most commonly used illegal/medicinal drug among Americans [1]. Marijuana is used recreationally to produce a pleasant feeling; however, it can cause short-term and long-term unwanted side effects. A few short-term impacts are altered sensations, hindered mobility, trouble thinking and solving problems, and memory issues. When taken in large doses, hallucinations, delusions, and psychosis may occur. Long-term effects may vary from physical to mental. Physical long-term effects include breathing problems since marijuana smoke irritates the lungs and causes increased heart rate and problems with child development during and after pregnancy. Mental long-term effects include cognitive impairment, anxiety, and paranoia [2]. Despite this, the use of cannabis for medicinal purposes is gaining prominence.

Although cannabis use is predominantly prohibited globally, laws governing it are becoming increasingly less stringent. Worldwide, decriminalization—which lessens punishments for personal use but not for distribution—is becoming more popular. This practice is seen in the Netherlands, Portugal, and some regions of Australia. Additionally, Peru, Germany, New Zealand, and the Netherlands, along with numerous U.S. states, have legalized medical use of cannabis. The only three nations that have nationalized the use of cannabis for recreational purposes are Canada, Uruguay, and Malta [3]. Cannabis possession is prohibited in the U.S. unless it is intended for authorized research purposes [4]. However, individual U.S. states started legalizing marijuana for recreational use in 2012, and by 2023, nearly half of all U.S. states have followed suit [3].

Though cannabis was only introduced into Western medicine around 180 years ago, it has been used for therapeutic applications for at least 3000 years. Cannabis was marketed for its analgesic, sedative, anti-inflammatory, antispasmodic, and anticonvulsant effects. The U.S. Treasury Department implemented the Marijuana Tax Act in 1937, which enforced a specific tax of $1 an ounce when cannabis was used for medical purposes and $100 an ounce when used for nonmedical purposes. This led to the drug being prescribed only in critical cases, and it was thought this would help prevent addiction. Then, the Controlled Substances Act was passed by Congress in 1970, designating marijuana as a Schedule I drug with a high potential for abuse or addiction and no FDA-approved medicinal use. However, when the Compassionate Use Investigational New Drug program was put in place in 1978, patients were given cannabis in critical cases, depending on the necessity [5].

The primary psychoactive substance in cannabis, Δ-9-tetrahydrocannabinol (THC), is responsible for many of the strong effects that users seek. THC comes from the resin produced by the leaves and buds of the flowers of the female cannabis plant. Cannabidiol (CBD) is another common cannabinoid (Figure 1), but it produces mild to no psychoactive effects compared to THC [1]. THC affects the body variously, depending on how it is consumed. When inhaled, THC is released into the bloodstream from the lungs and quickly travels to other areas of the body, including the brain. This leads to calming, a sense of ecstasy, relaxation, amplified sensory perception, laughter, altered perception of time, and increased appetite. When ingested through foods or drinks, the effects are delayed from 30 min to 1 h as the drug passes through the digestive system into the blood. A considerably lower amount of THC enters the bloodstream in a given period when eaten or drunk rather than smoked. This can lead people to consume a greater quantity of the chemical as concentrations gradually increase, causing some individuals unwanted effects of anxiety, fear, distrust, or panic [2].

The brain recognizes THC because its chemical structure allows it to bind to and activate receptors for anandamide, a natural chemical used for intercellular communication. This allows THC to alter brain and gut signaling. Very little is known about the effects produced by other cannabinoids, and the same kinds of detailed studies will be needed to characterize these to the extent that THC has been characterized. It is likely that these other cannabinoids can act additively or possibly synergistically with THC, so there is hope that research in this area will improve the current therapeutics for the whole cannabinoid family [1]. THC acts on cannabinoid receptors on neurons in several brain regions and modulates numerous essential tasks in the body, leading to many of the effects previously mentioned. Cannabinoid receptors have been found on most other tissues so far examined in our bodies, but their effects are just now being explored, together with the molecular mechanisms and signaling pathways that produce these effects. It is also unclear the full extent cannabinoids play in tissue development and repair, so there is also promise for new discoveries to be made in these areas [5].

Preclinical studies will need further validation in early-stage placebo-controlled clinical studies testing the effectiveness of cannabinoids for tumor growth suppression and antimetastatic effects.

## 2. Classification of Cannabinoids

The main categories of cannabinoids mentioned are endogenous cannabinoids (endocannabinoids) and phytocannabinoids. In the current review, we use the broad term cannabinoids to refer to a small number of naturally occurring compounds (THC and CBD) as well as a few synthetic analogs (see below), although, in a limited number of cases, minor differences between cannabinoids can occur.

Endogenous cannabinoids, compounds that activate cannabinoid receptors but are made by our bodies (e.g., anandamide), have been found to help with the response to pain, mood, inflammation, memory, movement, and others. Two key endogenous cannabinoids have been identified: anandamide (AEA or arachidonoyl ethanolamide) and 2-arachidonoyl glycerol (2-AG). These send chemical messages between neurons, serving as neurotransmitters. Endogenous cannabinoids affect the areas of the brain that control happiness, memory, reasoning, coordination, attentiveness, movement, and sensory and time perception [5].

2-arachidonoyl glycerol (2-AG) and arachidonoyl ethanolamide (anandamide) are the most researched endogenous cannabinoids. Although the structures of these two endocannabinoids are fairly similar, they have quite distinct physiological and pathological functions, given that they are produced and broken down by different enzymatic pathways [6].

Phytocannabinoids are essentially cannabinoids that occur naturally in the cannabis plant. Some blooming plants, liverworts, and fungi contain these bioactive natural substances that have the potential to treat human illnesses. The most popular, productive, and well-studied source of phytocannabinoids is *C. sativa* [7].

Synthetic cannabinoids, also known as herbal or liquid incense, are human-made mind-altering chemicals that are offered as liquids to be vaporized and inhaled using devices such as e-cigarettes. They can also be sprayed on dried, shredded plant material to be smoked. They were initially designed for therapeutic purposes; however, they are now more well-known for recreational purposes. There have not been many studies conducted on the effects of synthetic cannabinoids on the human brain. However, it is known that some have a greater affinity than marijuana for the cell receptors affected by THC and can have far more potent effects. Compared to the other two cannabinoids previously discussed, the consequent health effects might be more unpredictable and harmful [8].

## 3. Palliative Cannabis-Based Treatments for Cancer

Researchers had begun studying the possible therapeutic uses of cannabinoids in treating the side effects of therapy and tumor-related symptoms in the 1970s [9]. CBD has been found to have various valuable effects on cancer, including suppressing malignant cells, alleviating cancer-associated pain, and decreasing the unfavorable effects of chemotherapy, such as nausea and vomiting [10]. However, patient interest in using cannabinoids is also to be considered. In a retrospective study with 163 patients, Raghunathan et al. found that although they are aware of the side effects and need for solid evidence, cancer patients turn towards medical cannabis for treatment. Survey results showed that medical cannabis was usually used to promote sleep (53%), relieve pain (47%) and anxiety (46%), and improve appetite (46%). Surprisingly, 29% were interested in trying cannabinoid therapy as a treatment for cancer [11]. Similarly, a survey conducted by Weiss et al. to look into cannabis use among 612 breast cancer patients reported that though many patients use cannabis to treat cancer-associated symptoms, most do not discuss it with their doctors and rather obtain information from sources on the internet, family, and friends. 42% reported using cannabis for the relief of symptoms, including pain (78%), insomnia (70%), anxiety (57%), stress (51%), and nausea and vomiting (46%). A total of 75% reported that it was very/extremely helpful in treating their symptoms. About half of the respondents noted that cannabis would soon be used to treat cancer. Furthermore, when asked about their interest in using cannabis for medical uses, 64% responded that they were very/extremely interested, 23% responded that they were somewhat interested, and 13% were not/minimally interested [12]. To learn more about cannabinoids and their possible uses in cancer treatment, clinical trials and studies are being conducted, and more are planned.

Anecdotal success stories of cannabinoids are the main reason for the rise in the medical prescription of cannabis-derived oils and herbal cannabis. Oelen et al. surveyed 152 people who were aged 18 years or older and receiving intravenous systemic treatment at an outpatient facility in the Netherlands for solid cancer in order to gauge patient views towards the use of cannabinoids in medical treatment. Intravenous systemic therapy could be the sole treatment or a (neo) adjuvant therapy. A total of 65% of the 152 subjects received treatment with a palliative goal, while more than 40% (65 subjects) received immunotherapy. A variety of cancers were represented, with the majority being lung-related. The questionnaire included questions on cannabis consumption, sociodemographic factors, and clinical characteristics. Depending on their responses, patients were divided into one of the following categories: (1) never used cannabinoids, (2) recreative use of cannabinoids in the past, (3) medical use of cannabinoids in the past, (4) current recreative use of cannabinoids, and (5) current medical use of cannabinoids. Then, the frequency of cannabinoid consumption was calculated for each group. According to the report, nearly 25% of participants have used cannabinoids for medicinal purposes. About 23% of current nonusers reported possible future medical use. About 50% of users reported the potential of cannabinoids in cancer treatment as their primary reason for using these drugs. Overall, when compared to the general population, cancer patients consumed substantially higher amounts of cannabinoids [13].

Yet, in a systematic meta-analysis reviewing 17 research articles, Belgers et al. discovered that cannabinoids had no effect on health-related quality of life (HRQoL) or mental well-being. However, just two studies have looked into how cannabis affects the HRQoL in patients with brain tumors. One study found no improvement in HRQoL when THC was used to treat chemotherapy-induced nausea and vomiting in 32 patients with primary brain tumors. The second study reported that there was no difference in HRQoL between different CBD:THC ratios when it looked at two different CBD:THC ratios (1:1 and 1:4) and their impact on HRQoL in 88 patients with primary malignant brain tumors, but neither investigation reported significant effectiveness [14].

Interestingly, a case-control study performed by Shewale et al. at Ohio State University found that long-term marijuana use might act as a protective factor against HPV-negative cancers. The analysis found that regular marijuana use for many years independently lowered the likelihood of HPV-negative cancers. It was mentioned that Δ-9-tetrahydrocannabinol and cannabinol could potentially protect against oral cancer through immunomodulatory effects. However, there was no evidence indicating a relation between marijuana use and HPV-positive cancers [15].

Though there have been several studies in adult oncology, there are only a few studies on the role of cannabinoids in childhood cancers. Schab and Skoczen found that cannabinoids hindered the growth of rhabdomyosarcoma and neuroblastoma cells and stopped cell cycle progression in osteosarcoma cells in children. Moreover, they reported that CB1 expression may help foreshadow the shrinking of low-grade gliomas in children [16]. Another review on the role of cannabinoids in pediatric oncology indicated that a majority of previous studies were based on acute lymphoblastic leukemia and suggested that cannabinoids have the ability to eliminate cancer cells both in vivo and in vitro [17].

So far, cannabinoids have been proven to be safe in treating cancer but not useful in controlling or curing the disease [18]. Cannabinoids have been found to have an effect on tumors of the brain, prostate, colon and rectum, breast, uterus, cervix, thyroid, skin, pancreas, and lymph [9]. THC has been found to help mitigate pain, decrease inflammation, and act as an antioxidant [18]. However, the full potential of cannabinoids is yet to be understood. Few drugs, including Dronabinol (Marinol^®^), Nabilone (Cesamet^®^), and Nabiximols, which contain a mix of CBD and THC, have been approved for use in the USA for cases where other treatments have failed. Dronabinol, which is usually in the form of a capsule, is used to help relieve chemotherapy-induced nausea and vomiting, but it appears to be about equally as effective as megestrol acetate, a progestin medication. Nabilone, usually taken by mouth, is a synthetic cannabinoid similar to THC and is also used to help relieve chemotherapy-induced nausea and vomiting. Nabiximols, usually taken in the form of a mouth spray containing Δ-9-THC and CBD in a 1:1 ratio, are still under study in the United States but have promising potential for relieving cancer-associated pain, though some studies have not shown positive results. According to the American Cancer Society, further scientific research on cannabinoids for cancer patients is necessary. Additionally, more advanced and efficient medicines are required to combat the complications of cancer and its treatments [18].

## 4. Clinical Studies on the Use in Chemotherapy-Induced Nausea and Vomiting

The FDA-approved cannabinoids dronabinol and nabilone are suggested to be used to treat nausea and vomiting that is unresponsive to traditional antiemetic medications by the American Society of Clinical Oncology Focused Guideline. However, the guidelines do mention that there is still inadequate evidence to make a recommendation on the use of medicinal marijuana to treat nausea and vomiting in cancer patients undergoing chemotherapy or radiation therapy [19]. Numerous studies are looking into how cannabinoids affect chemotherapy-induced nausea and vomiting (CINV).

There are three cannabinoids accessible through the pharmaceutical market right now. Taylor et al. reviewed the pharmacokinetics, side effects, toxicity, dosing, pharmacodynamics, and monitoring of cannabinoid antiemetics. Both Dronabinol and Nabilone, forms of synthetic THC, have been authorized by the FDA for treating a variety of illnesses, including CINV. A CBD medication was more recently given FDA approval to treat seizures and a few other disorders but has not yet been deemed safe to be used as an antiemetic [20]. Nabilone is usually offered as a 1 mg PO capsule. Nabilone’s initial dosage ranges from 1 to 2 mg, administered twice a day to treat refractory nausea and vomiting brought on by cancer treatment. Dosing can continue for up to 48 h following the last chemotherapy session. The maximum daily dose is 2 mg. Dronabinol comes in two forms: a 2.5 mg, 5 mg, or 10 mg PO pill or a 5 mg/mL PO solution using a calibrated syringe. A starting dose of 5 mg should be given 1 to 3 h before starting chemotherapy, and then every 2 to 4 h after, for a total of 4 to 6 doses every day. The first dose should ideally take place after a 10-hour fast and at least 30 min before a meal. The following doses do not have to be administered with regard to meals [21].

The emetic reflex is physiologically suppressed by CB1 receptors by preventing the release of excitatory transmitters. Notably, CB1R is present on dopaminergic, noradrenergic, and other neurons located in the parts of the brain that control nausea and vomiting, which are still not fully understood. The use of cannabinoids, particularly THC, must be carefully monitored and regulated given the potential side effects that patients may encounter, including psychotomimetic reactions, mood changes (dysphoria, euphoria, anxiety, or panic reactions in some new users), and some limited toxicity, mostly linked to synthetic cannabinoids [9].

Polito et al. conducted a multicenter, retrospective review of pediatric patients who received nabilone for acute CINV to assess its safety and effectiveness. A total of 110 eligible patients with a median age of 14 who were treated for CINV prophylaxis with nabilone and a 5-HT3 antagonist were recruited. In total, 34% (37/110) of the children who took nabilone reported experiencing adverse reactions. The most frequently documented side effects, though of minimal clinical relevance, were sedation (20.0%), dizziness (10.0%), and euphoria (3.6%). It was concluded that children taking nabilone as part of their antiemetic regimen had poor acute CIV control and that future research should concentrate on establishing CINV prophylaxis and treatment that is consistent with guidelines [22].

Between August and November 2022, Sukpiriyagu et al. performed a randomized, double-blinded, crossover, and placebo-controlled trial to assess how effective tetrahydrocannabinol (THC):cannabinoid (CBD) (1:1) oil is in lowering CINV. Gynecologic cancer patients who received moderate-to-high emetogenic chemotherapy were randomly assigned to one of two groups (A or B). Thai GPO produced the THC:CBD extract oil 1:1 (CX) used in this investigation. One mL of CX contained 2.7 mg of THC and 2.5 mg of CBD. To lessen the bias caused by interpersonal variation, cases were crossed over between placebo and THC:CBD. In the first cycle, group A received the 1:1 THC:CBD extract oil prior to receiving chemotherapy, and group B received the placebo. In the second cycle, group A received the placebo, while group B received the 1:1 THC:CBD extract oil. The nausea score and side effects were recorded. A total of 60 participants were recruited, and after exclusion, 54 participants were included in the study. Subjects who received CX had considerably lower average nausea scores than those who received a placebo (2.1 vs. 3, *p* = 0.001). The main side effects were sedation and dizziness. The researchers concluded that among patients with gynecologic cancer who underwent highly emetogenic chemotherapy, the cannabinoid extract (THC:CBD) was a suitable adjuvant treatment to minimize CINV [23].

Grimison et al. performed a multicenter, randomized, double-blinded, placebo-controlled trial among 72 participants experiencing CINV during moderate-to-high emetogenic intravenous chemotherapy despite guideline-consistent antiemetic prophylaxis. Participants were given one cycle of 1–4 self-titrated oral THC 2.5 mg/CBD 2.5 mg (TN-TC11M) capsules three times per day from days 1 to 5, as well as one cycle of a placebo in a crossover design. A third cycle blinded patient preference. THC:CBD increased the complete response from 14% to 25%, with a similar impact on preventing emesis and substantial nausea. Despite the fact that 31% of individuals reported moderate to severe cannabinoid-related side effects such as sedation, dizziness, or disorientation, 83% of respondents preferred cannabis to the placebo. There were no reported severe side effects of THC:CBD [24].

## 5. Clinical Studies on the Use in Cancer-Associated Pain

Many cancer patients rely on opioids for pain relief but face a multitude of dangerous side effects as a result. The earliest uses of cannabinoids in oncology were to treat pain resulting from treatment and the disease itself. This led to cannabinoids replacing opioids in treatment [9]. For example, an oromucosal spray containing THC and CBD is being clinically used in cancer treatment if opioid treatment is not effective. Studies have shown that it is effective for prolonged use and does not produce drug resistance [10].

Meng et al. found that, on average, patients already receiving opioids for advanced cancer pain exhibited lower pain scores when treated with nabiximols. In a large observational study of cancer patients using cannabis, researchers followed patients over a period of 6 months, and data showed that more patients reported increased quality of life and decreased opioid use. Fewer patients reported severe pain. Other recent findings that were mentioned include a three-week cohort study on employing nabiximols for advanced cancer pain in patients already maximized on opioids, in which researchers reported an improved average pain score. It was determined that strong preclinical animal data and a substantial amount of observational evidence support the possibility of cannabis being useful in the management of cancer pain. However, clinical efficacy is only weakly supported by clinical trial data. Therefore, high-quality randomized controlled trials are still required to accurately compare the efficacy and safety of medical cannabis to placebo and conventional therapies for cancer-related symptoms [25].

Schleider et al. reviewed data that were regularly gathered during a program of 2970 cancer patients treated with medical cannabis between 2015 and 2017. The mean age of the participants was 59.5 ± 16.3 years. Women made up 54.6% of the population. A total of 26.7% of the patients said they had previously used cannabis. This study’s objectives were to investigate the safety and effectiveness of medicinal cannabis therapy and to describe the epidemiology of cancer patients receiving it. Breast (20.7%), lung (13.6%), pancreatic (8.1%), and colorectal (7.9%) cancers were the most prevalent, with stage 4 cancer accounting for 51.2% of cases. Sleep issues (78.4%), pain (77.7%, median intensity 8/10), weakness (72.7%), nausea (64.6%), and loss of appetite (48.9%) were the main side effects reported. A total of 902 patients (24.9%) died, and 682 (18.8%) discontinued treatment after six months of follow-up. Of the remaining patients, 1211 (60.6%) provided a response; 95.9% said their condition had improved, 45 (3.7%) said their condition had not changed, and four (0.3%) said their condition had become worse. Based on this information, the researchers came to the conclusion that cannabis appears to be a well-tolerated, efficient, and safe option for patients to manage pain and other symptoms associated with cancer [26].

A total of 25 patients with advanced cancer who were experiencing uncontrollable pain were recruited for a two-stage study conducted by Clarke et al. to test the efficacy of an orobuccal spray containing nanoparticles of 9-tetrahydrocannabinol (THC) and cannabidiol (CBD). Stage I was a simple two-day pharmacokinetic investigation with a single ascending dose (SAD) and a multiple ascending dose (MAD) of the nanoparticle Δ-9-THC/CBD formulation. Participants were then invited to participate in Stage II if eligible. The 30-day period was divided into three phases: a dose escalation phase during days 1 to 9, a treatment phase during days 10 to 15, and a follow-up phase during days 16 to 30. The MDCNB-01 formulation, which was made from plant material, had 97.5% THC and CBD and 2.5% cannabidiolic acid, cannabigerol, cannabinol, and cannabichromene. The daily pain NPRS score was determined by taking the average of all daily assessments. The greatest mean pain score improvement from baseline was observed in patients with breast and prostate cancer, at 40%. Mild or moderate drowsiness, which affected 44% of patients, and nausea and vomiting, which affected 72% of patients, were the most frequently reported side effects. Overall, the researchers concluded that the medicine showed early signs of analgesic effectiveness and was safe and tolerated in patients with advanced, incurable malignancies [27].

## 6. Antitumor Properties of Cannabinoids

Researchers have been interested in the therapeutic uses of cannabinoids in cancer treatment due to several beneficial properties. Cannabinoid agonists can regulate cancer-related cell cycle pathways by binding to CB1 or CB2 receptors [9]. This produces many desirable effects in cancer therapy, including tumor growth prevention, cell cycle obstruction, and cell death [7]. Cannabinoids have been found to work independently of some receptors but dependently on other receptors, like TRPV1 [9]. An excellent review of TRPV1 involvement, together with other receptors like GPRs, has recently provided a schematic diagram of receptor combinations and the complex signaling pathways involved [28].

Cannabinoids may also cause apoptosis by activating the MAPK pathways (which promote cellular proliferation, differentiation, development, inflammatory responses, and apoptosis) and inhibiting the PI3K-Akt pathway (which promotes metabolism, proliferation, cell survival, growth, and angiogenesis) [9]. Cannabinoids also appear to affect the MEK-extracellular signal-regulated kinase signaling cascade and the cyclic AMP-protein kinase-A pathway [29]. Multiple studies have shown that treatment with THC and CBD inhibits metastasis [9].

Preclinical research has offered encouraging results that show that sativa extracts with high ratios of THC:CBD, THC:CBG, or THC:CBN exhibit anticancer activities against squamous cell carcinoma growth. Li et al. found varying levels of effectiveness of these extracts on HCC1806 squamous cell carcinoma cells, from 66% to 92% of growth inhibition. They conducted clustering and association analysis to determine the relationship between the chemical makeup of THC, CBD, CBG, and CBN, and twenty major terpenes and their effectiveness in preventing the proliferation of cancer and reducing inflammation. Terpinene’s presence was shown to be positively correlated (*p*-value = 0.002) with anticancer activity; eucalyptol was second with a *p*-value of 0.094. Camphor had a negative correlation with the suppression of IL6 expression, while P-cymene and β-myrcene had a positive correlation. Following that, they conducted a correlation analysis between terpenes and cannabinoids and discovered that there was a positive association for the following pairs: α-pinene vs. CBD, p-cymene vs. CBGA, terpenolene vs. CBGA, and isopulegol vs. CBGA. They also demonstrated varying degrees of anti-inflammatory activity. They concluded that while only a few specific high-THC extracts exhibited anti-inflammatory action, the majority of them did exhibit anticancer activity [30].

Loubaki et al. reported similar effects on oral cancer cells. An MTT experiment was conducted to evaluate the effects of various concentrations of an 8-component cannabinoid mixture (CM) on the proliferation of Ca9-22 oral cancer cells and GMSM-K gingival normal cells. When treated with different concentrations (0, 0.1, 1, and 2 μg/mL) of the cannabinoids mixture to study their growth, Ca9-22 oral cancer cells faced high cytotoxicity. In response to exposure to doses of 1 g/mL, levels of apoptosis, autophagy, antioxidants, mitochondrial stress, and DNA damage rose. This suggests that at a concentration higher than 0.1 µg/mL, cannabinoids may have a positive effect on treating oral cancer. The researchers concluded that treating oral cancer cells with this mixture at low concentrations (0.1 to 1 µg/mL) induced selective cell death and, as a result, emphasized the anticancer properties of cannabinoid derivatives [31].

Kiskova et al. found that THC and CBD obstructed the growth of the disease in breast cancer models, suggesting that cannabinoids may slow down tumor growth in breast cancer patients. The cannabinoids used G-protein-coupled CB-receptors (CB-Rs), CB1-R, and CB2-R, and various other receptors to affect the signaling pathways of cells. They block critical active pro-oncogenic signaling pathways (for example, the extracellular-signal-regulated kinase pathway) to inhibit cell cycle progression and cell growth and induce cancer cell apoptosis. The cannabinoids were found to be active in both estrogen receptor-positive and estrogen receptor-negative breast cancer cells. Cannabinoids were previously administered to patients with advanced stages of breast cancer, but Kiskova et al. suggested that they might be more beneficial if given at earlier stages [32].

More recently, Oliveira et al. examined the effects of somatostatin (SST) (a growth hormone inhibitory peptide that exists in two naturally occurring bioactive isoforms), cannabidiol (CBD), alone or with both SST and CBD, on receptor expression (CBR1 and CBR2, SSTR_2_ and SSTR_5_), cell proliferation and apoptosis, and associated downstream signaling pathways in MDA-MB-231 and MCF-7 breast cancer cells. MCF-7 cells treated with SST alone or in conjunction with CBD demonstrated inhibition of phosphorylated Protein Kinase B (pAKT) and phosphorylated-Phosphoinositide 3-Kinase (pPI3K) expression when compared with MDA-MB-231 cells, indicating that they exhibit changes in cell proliferation, apoptosis, and signaling pathways in response to receptor activation in a cell-type-dependent manner [33].

Gómez del Pulgar et al. also found THC and other cannabinoids to induce apoptosis of cancer cells, specifically glioma cells, by producing ceramide. They suggested that malignant gliomas can be treated with cannabinoid therapy because they selectively induce ceramide synthesis and apoptosis in transformed glial cells but not in nontransformed neural cells. Primary neurons and astrocytes are not affected by cannabinoids, as they do not lead to apoptosis or ceramide accumulation [34].

Many studies have also supported the idea that cannabinoids affect cancer-related signaling pathways. For example, Zaiachuk et al. reported in a recent review that the endocannabinoid system has been altered in colorectal cancer and that cannabinoids have been reported to affect important signaling cascades involved in cancer development, including pathways affected by immunotherapy. The researchers further explained that certain in vitro experiments have shown that CBD inhibited colorectal cancer cell proliferation via CB1, TRPV1, and PPAR receptors. In vivo models of azoxymethane-induced colon cancer confirmed the chemoprevention of CBD. The amount of 1 mg/kg of CBD decreased the proliferation of epithelial cells located in colonic crypts by 67%, tumor count by 66%, and polyps by 57%. THC has also been demonstrated to cause apoptosis in colorectal cancer cells by activating the CB1 receptors and BAD and inhibiting PI3K-AKT and the RAS-MAPK cascade [35]. Another review by Mangal et al. similarly mentioned that “increasing data from and in vivo studies have started to show evidence of cannabis in modulating signaling pathways involved in cancer cell proliferation, autophagy, apoptosis and inhibition of angiogenesis and metastasis”. It was further added that the antitumor effects of cannabinoids tend to modulate processes like apoptosis and autophagy by first promoting ceramide synthesis, which leads to the activation of ER stress-related signaling proteins. This results in the inhibition of the AKT/mTORC1 axis, which causes cell cycle arrest and other processes, like cell death and aging. The stimulation of MAPK/ERK signaling through calcium induction is another mechanism of cannabinoids that can potentially affect cancer-related signaling pathways. The researchers concluded that “strategies that would optimize the anticancer effects of cannabinoids through interference of these signaling cross-talks may prove useful for therapeutic intervention” [36]. A schematic diagram that summarizes these rather complex pathways was provided in a recent review. Peeri and Koltai claimed that “specific combinations of multiple phytocannabinoids act synergistically against cancer cells and may trigger different anticancer signaling pathways”. The researchers mention evidence that the activation of cannabinoid receptors modifies the expression of regulatory genes involved in stem cell proliferation and differentiation, which hinders the invasiveness and tumorigenesis of glioblastoma stem cells. CBD was found to increase reactive oxygen species (ROS) and activate the p38-MAPK signaling pathway. These effects resulted in the downregulation of important stem cell regulators and the inhibition of the survival and self-renewal of cultured primary GSC [37]. The data from these sources, among many others, support that cannabinoids may be used to supplement commonly used therapies for treating cancer.

Some cannabis chemovars were found to display more effective antitumor properties on some types of cancers compared to others. Some chemovars were more successful in treating cancer in general, while some were greatly successful in treating one specific cancer. Though more prevalent in higher percentages of THC, this did not have a significant impact on the antitumor abilities of the drug. Whole extracts frequently performed better than the applications of THC by itself. Medicinal extracts of cannabis and sole cannabinoid administration have shown variable efficacy in the treatment of several cancer types. For example, melanoma cell lines were found to have lower functionality when treated with a combination of CBD and THC, whereas they performed better when treated with only THC. The functionality of cervical cancer cell lines was also found to be significantly lower when exposed to CBD, compared to whole cannabis extracts, displaying higher occurrences of cell death. Further, higher levels of Δ-9-THC cannabis extracts have been shown to boost cell death in PC-3 prostate adenocarcinoma and malignant melanoma cell lines. However, it is to be noted that the effectiveness and response to cannabis product treatment are impacted differently by complex interactions between terpenes and other minor cannabinoids. The effects of the endocannabinoid system’s involvement in the proliferation, metastasis, angiogenesis, and tumorigenesis of cancer may serve as important sources of inspiration for the creation of novel cancer treatment methods [38]. Blal et al. examined the antitumoral effects of 24 cannabis extracts representative of three primary types of chemovars on head and neck squamous cell carcinoma (HNSCC). Extract 296 (CAN296) was the most potent in inducing HNSCC cell death via proapoptotic and antiproliferative effects, with the CBD fraction as the primary inducer of the antitumoral activity. Cells were assessed via WST-1 proliferation assay. The cannabis extract had a significant antiproliferative effect on Scc25 at 4–6 µg/mL when applied for 72 h. At 8 µg/mL, the extract was already effective after 24 h. It was suggested to use decarboxylated CBD-type extracts enriched with CBC for future preclinical trials aimed at HNSCC treatment [39].

According to growing data from preclinical research, tetrahydrocannabinols (THCs) limit metastasis and angiogenesis, two important features of cancer pathogenesis, while promoting apoptosis and autophagy. However, the origin of the tumor site, the expression of cannabinoid receptors on tumor cells, and the types and quantities of THC all affect how much of an anticancer effect they have. CB1 and CB2 expression is typically substantially higher in tumor cells. THCs work by activating the CB1 and CB2 cannabinoid receptors, whose communication with the endocannabinoid system is crucial for regulating both euphoric effects and cell survival and proliferation [40]. The in vitro studies described above have started to unravel some of the molecular and signaling pathways involved in these interesting new cannabinoid effects, but much work remains to be conducted to characterize their biochemical effects. These preclinical studies will need further validation in early-stage placebo-controlled clinical studies testing the effectiveness of cannabinoids for tumor growth suppression and antimetastatic effects.

While these preclinical results are very encouraging, it is clearly time to examine cannabinoid antitumor actions in real-world, well-controlled, in vivo animal studies and, when possible, in early-stage clinical trials. A small clinical trial showed promising results in glioblastoma patients who were unresponsive to other interventions. THC was directly injected into the tumor mass in two patients and produced a reduction in tumor size. The authors concluded that: “The fair safety profile of THC, together with its possible antiproliferative action on tumor cells reported here and in other studies, may set the basis for future trials aimed at evaluating the potential antitumoral activity of cannabinoids.” [41]. Several new clinical trials are underway that will provide better indications of cannabinoid therapeutics (e.g., NCT05520294, NCT05629702, NCT01489826, NCT03529448, and NCT01812603). It should be possible to mine earlier clinical datasets where cannabinoids have been used to look at tumor remissions even though these were not the primary endpoint in the initial studies.

## 7. Conclusions

The possible therapeutic properties of marijuana have been up for debate and research for a long time. Some THC-based medications, including dronabinol (Marinol^®^) and nabilone (Cesamet^®^), have already been approved for use in pill form by the FDA for treating chemotherapy-induced nausea. Various other cannabinoid medications are currently being studied or approved for use in treating cancer or related symptoms. However, the impact of long-term use on those with health- and/or age-related vulnerabilities is not well understood. Clearly, it will be necessary to conduct more research to advance our understanding of the various uses of medical marijuana [2]. Given the recent findings, it will be most interesting to see whether synergistic effects occur with cannabinoids and traditional chemotherapeutics and analgesics for cancer treatment. Early-stage clinical trials investigating the ability of cannabinoids to suppress tumor growth and/or metastasis are needed. Studies are also needed to examine possible synergistic/additive effects of cannabinoids used in combination with current anticancer therapies.

## Figures and Tables

**Figure 1 ijms-25-00074-f001:**
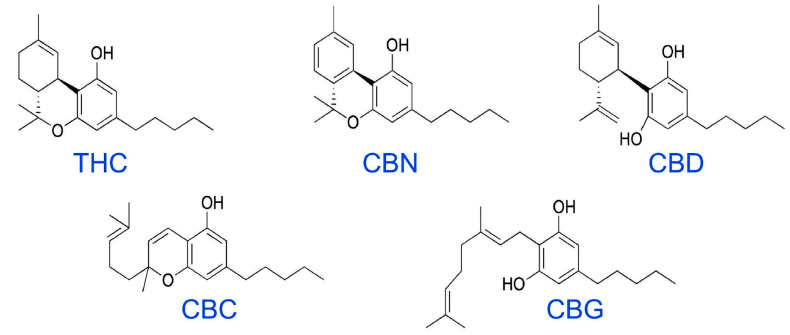
Over 50 cannabinoid molecules have been extracted from marijuana plants, and several have been shown to have pharmacologic effects. The five molecules shown here are the most common cannabinoids available for use in humans: THC—Δ-9-tetrahydrocannabinol; CBN—cannabinol; CBD—cannabidiol; CBC—cannabichromene; and CBG—cannabigerol. It should be noted that most cannabinoids have not been characterized nor investigated for pharmacologic activity.

## Data Availability

Not applicable.

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
