# Peer review of "Cannabinoids in Treating Chemotherapy-Induced Nausea and Vomiting, Cancer-Associated Pain, and Tumor Growth"

_ijms, 2023, doi:10.3390/ijms25010074_

Round 1

Reviewer 1 Report (New Reviewer)

Comments and Suggestions for Authors

The paper titled Cannabinoids in Treating Chemotherapy-Induced Nausea and Vomiting, Cancer-Associated Pain, and Tumor Growth is a very well-written review containing information about the cannabinoids roles in some cancer-related processes such as emesis, pain and cell proliferation.

I recommend this review for publication after revision.   This review includes recent information regarding the use of some cannabinoids to counteract the emetic and painful effects that accompany chemotherapy. Additionally, this review has some information about the impact of these compounds on some cancer cell lines. The authors could consider the following suggestions to clarify some concepts for the readers.   1.- Section 6: 6. Antitumor Properties of Cannabinoids Please consider adding more information about the Cannabinoids antitumor effects through TRP channels and GPR55. (for example: https://doi.org/10.1111/cas.15257)

Please clarify: How do cannabinoids produce apoptosis if they activate MAPK pathways associated with cellular proliferation? It is confusing to the reader. This information is in the following paragraph:

….The following paragraph: They may cause apoptosis by activating 333 the MAPK pathways (which promote cellular proliferation, differentiation, development, 334 inflammatory responses, and apoptosis) and inhibiting the PI3K-Akt pathway (which pro- motes metabolism, proliferation, cell survival, growth, and angiogenesis).Cannabinoids also appear to affect the MEK-extracellular signal-regulated kinase signaling cascade and  the cyclic AMP-protein kinase-A pathway.5 Multiple studies have shown that treatment 338 with THC and CBD inhibits metastasis….. 2.- Please clarify in the following paragraph which receptor modifies its expression with the treatment with SST and CBD? …..More recently, Oliveira et al. examined the effects of somatostatin (SST) (a growth 377 hormone inhibitory peptide that exists in two naturally occurring bioactive isoforms), cannabidiol (CBD), alone or in combination, on receptor expression (WHICH RECEPTOR IS?), cell proliferation and 379 apoptosis, and associated downstream signaling pathways in MDA-MB-231 and MCF-7 breast cancer cells…….. 3. please clarify how CBD inhibit colorectal cancer cell proliferation through

CB1, TRPV1, and PPAR- receptors (lanes 396-398).

4.-  What do AOM and ACF mean in lanes 398-399? Please clarify between brackets.

5. How does THC inhibit CB receptors? (lanes 456), please explain.

Finally, I suggest that authors add a figure to schematize the main cell pathways for the THC and CBD effects on cell proliferation.

Author Response

Thank you for your help improving our paper.

We have incorporated all of your suggestions.

Point by point responses are provided in the attached word document.

Reviewer 2 Report (New Reviewer)

Comments and Suggestions for Authors

Cannabinoids in Treating Chemotherapy-Induced Nausea and Vomiting, Cancer-Associated Pain, and Tumor Growth

Authors: Pavana P Bathula and M Bruce Maciver

At the outset, this paper discusses an important topic and is well written. Other esteemed reviewers have done an excellent job in reviewing it and have suggested very valid suggestions.  Below are a few additional suggestions.

It is important to remember that Cannabis sativa, Linn., plant has 125 cannabinoids of which only two-THC and CBD have been extensively studied for pharmacological activity/medical value with multiple clinical trials underway. However, as of today, only four cannabis products are approved for medicinal use: (i) synthetic THC (dronabinol, Marinol) for treating chemotherapy-associated nausea and vomiting (but not pain); (ii) a synthetic cannabinoid, nabilone (Cesamet), for treating chemotherapy-associated nausea and vomiting (also not for cancer pain) and for stimulating appetite in HIV-infected patients; (iii) CBD (Epidiolex) for treating two rare forms of seizures: (a) Lennox-Gastaut and Dravet syndromes in young children, and (b) tuberous sclerosis from a benign brain tumor; (iv) CBD plus THC in 1:1 ratio (Sativex) in 28 countries (not in the US) for treating multiple-sclerosis-associated spasticity/neuropathic pain. Therefore, it is of paramount importance that the authors should be careful in giving the impression that cannabinoids (and to be specific which cannabinoid) can be used to treat cancer pain. Even Sativex (CBD-THC in 1:1 ratio) is approved (in 28 countries but not in the US) for treating only MS-associated neuropathic pain and not any other pain.   

In this reviewer’s opinion, the authors should consider clearly stating in their summary or conclusion that currently neither cannabis nor any of its compounds including THC, CBD or a combination thereof is approved by any regulatory agency for treating cancer and/or non-cancer pain. pain. Furthermore, there are insufficient data from clinical studies to support the use of any cannabinoid for treating any cancer/tumor growth. However, there are a few clinical trials registered at https://clinicaltrials.gov to determine if CBD, THC and/or THC+CBD can be used to treat cancer pain.

Finally, the authors should be specific when they use the term ‘cannabinoids’. They should specify which cannabinoids they are talking about. Is it THC or CBD or any other.  Currently, at least 6 additional cannabinoids including cannabichromene, cannabinol, cannabigerol, cannabidivarin, tetrahydrocannabivarin, cannabicyclol, have been studied for pharmacological activity but with very limited success in determining their medicinal value.  Therefore, this reviewer urges authors to be more specific when describing or discussing medicinal value of cannabis compounds. It is also urged to not give impression that cannabis plant used in any form, whether smoked or ingested, is medicinal, even though, in the absence of clear clinical efficacy data, many states in the US and other countries have approved it as medicine. The US FDA or any other regulatory body has not approved it as medicine.   

It is hoped that the authors will take these constructive suggestions.

Recommendation: Accept with minor revision.

Author Response

Thank you for your kind words and helpful suggestions.  We have adopted all suggestions and incorporated them in our improved manuscript.  Detailed responses are provided in the attached word document.

This manuscript is a resubmission of an earlier submission. The following is a list of the peer review reports and author responses from that submission.

Round 1

Reviewer 1 Report

Comments and Suggestions for Authors

The authors didn't adequately fulfill my concerns. 

Reviewer 2 Report

Comments and Suggestions for Authors

The topic of the review on Cannabinoids in Treating Chemotherapy-Induced Nausea and Vomiting, Cancer-Associated Pain, and Tumor Growth is of interest. However, the manuscript needs to be better organized (see the comments).

“The brain recognizes THC because its chemical structure is similar to anandamide, a 66 natural chemical used for intercellular communication (see Figure 1).” THC and anandamide doesn’t have similar structure. Moreover, they bind differently in the cannabinoid receptors. This sentence needs to be removed. Anandamide and 2-AG structures should be shown in Figure 1 since they are mentioned in the text.

The topic of the review on Cannabinoids in Treating Chemotherapy-Induced Nausea and Vomiting, Cancer-Associated Pain, and Tumor Growth is of interest. However, the manuscript needs to be better organized (see the comments).

Endogenous cannabinoids, compounds that are similar to cannabinoids…” Endogenous cannabinoids are cannabinoids. Are the authors refering to phytocannabinoids? To synthetic cannabinoids? The sentence doesn’t have any meaning. Endogenous cannabinoids are not similar structurally to phytocannabinoids. Moreover, they modulate the cannabinoid receptors somehow as THC does but not as does CBD or other phytocannabinoids.

“…have been found to help in the response to pain, mood, inflammation, memory, and movement”. The physiological and pathological responses of endocannabinoids are not limited to the ones reported by the authors.

Classification of cannabinoid: the synthetic cannabinoids are missing in this section since Nabilone, is a synthetic cannabinoid with therapeutic use as an antiemetic and as an adjunct analgesic for neuropathic pain used in clinical under the name Cesamet.

Preclinical research has offered encouraging results that show that sativa extracts with high ratios of THC:CBD, THC:CBG, or THC:CBN exhibit anti-cancer activities against squamous cell carcinoma growth.14” This study should be reported in the corresponding Antitumor Properties of Cannabinoids section and not in the present section.

Loubaki et al. reported similar effects on oral cancer cells.27 When treated with different concentrations of a cannabinoids mixture to study their growth, Ca9-22 oral cancer 280 cells faced high cytotoxicity.27 In response to exposure to doses of 1 g/mL, levels of apoptosis, autophagy, antioxidants, mitochondrial stress, and DNA damage rose.27 This suggests that at a concentration higher than 0.1 μg/mL, cannabinoids may have a positive effect on treating oral cancer.27” This study should be reported in the Antitumor Properties of Cannabinoids section and not in the present section.

In the section Clinical Studies on The Use in Chemotherapy-Induced Nausea and Vomiting, one is expecting human Clinical studies but not animal or cell studies that belong to pre-clinical stage. This needs to be better organized.

The reviewer agrees with “While these pre-clinical results are very encouraging, it is clearly time to examine cannabinoid anti-tumor actions in real-world, well controlled, in vivo animal studies and, when possible in early-stage clinical trials.”. However why didn’t the authors report the past and current clinical trials in this review?

Such as:

Guzman, M. et al. A pilot clinical study of Δ9-tetrahydrocannabinol in patients with recurrent glioblastoma multiforme. Br. J. Cancer 95, 197–203 (2006)

Or from a search in clinicaltrials.gov, some comments could be added to the review

https://clinicaltrials.gov/ct2/show/NCT05520294?term=cannabinoid&cond=Cancer&draw=2&rank=4

https://clinicaltrials.gov/ct2/show/NCT05629702?term=cannabinoid&cond=Cancer&draw=2&rank=5

https://clinicaltrials.gov/ct2/show/NCT01489826?term=cannabinoid&cond=Cancer&draw=2&rank=7

https://clinicaltrials.gov/ct2/show/NCT03529448?term=cannabinoid&cond=Cancer&draw=5&rank=31

https://beta.clinicaltrials.gov/study/NCT03529448?distance=50&cond=Cancer&term=THC&page=2&rank=12

https://beta.clinicaltrials.gov/study/NCT01812603?distance=50&cond=Cancer&term=THC&page=2&rank=19

etc….

Δ 9 –tetrahydrocannabinol should be Δ9-tetrahydrocannabinol

Reviewer 3 Report

Comments and Suggestions for Authors

The manuscript "Cannabinoids in Treating Chemotherapy-Induced Nausea and Vomiting, Cancer-Associated Pain, and Tumor Growth" by Bathula and Maciver reviews and discusses current knowledge on the potential application of cannabinoids for the treatment of cancer-related nausea and vomiting, pain, and tumor growth.

This is a timely topic, as there is increasing discussion on the potential therapeutic benefits of cannabinoids. However, this review seems to only address this topic superficially, as it does not provide sufficient details on the published data to support the conclusions withdrawn, thus showing a lack of a critical attitude towards the data found in the literature. Please note that clinical trials so far have not provided clear evidence of the potential therapeutic benefits of cannabinoids for the above-mentioned application. Moreover, it seems to provide a slightly biased perspective of the topic, as it mainly focuses on the potential therapeutic applications of cannabinoids. In fact, it would also be important to discuss the adverse effects commonly associated with using these substances, as these cannot be dissociated from their pharmacological action. Indeed, it should be noted that cannabis use has been associated with, for example, a 3.2-fold higher risk of developing neuropsy­chiatric disorders (e.g., psychosis) compared to individuals that had never used cannabis (Di Forti et al The Lancet Psychiatry, 6(5), 427-436).

These and other major shortcomings are detailed below:

1.      Lines 27-28:  besides mentioning the different cannabis strains, in the context of discussing the recreational/therapeutic use of this plant, it would be more helpful to distinguish cannabis in terms of its chemovars, i.e., high THC, similar THC/CBD content, high CBD.

2.      Line 29: it would be important to discuss the unwanted side effects of cannabis, as these should be taken into account while addressing its potential therapeutic effects.

3.      Lines 30-42: The authors focused only on US legislation. But what about other world regions (e.g., Europe, Asia, Africa)? Please have in mind that cannabis is the psychoactive substance most consumed worldwide, not only in the US. It should also be noted that cannabis is considered an illicit drug among countries that signed the 1961 United Nations Single Convention on Narcotic Drugs.

4.      Lines 66-88: It would be important to describe the pharmacodynamics of cannabinoids. Which receptors are targeted by cannabinoids? What is the result at cellular/tissue level of their activation/inhibition?

5.      The manuscript is mostly based on the description of trials data. Providing the molecular-based scientific foundations that possibly underlie the referred therapeutic effects would improve the quality of the manuscript by helping understand the potential therapeutic actions of cannabinoids, as well as the possible adverse effects that could be associated with the same targets.

6.      Lines 90-91: Mentioning only endogenous and phytocannabinoids is rather limiting. What about synthetic cannabinoids? These are most potent than endo- and phytocannabinoids and have actually been initially designed aiming at therapeutic applications.

7.      Line 116: what kind of intravenous systemic treatment were patients being administered?

8.      The authors do not provide details on the clinical trials (e.g., number of patients involved, the quality of the data, use of placebo controls, treatment randomization, risk bias assessment, possible confounding factors) that would be important to ascertain the evidence level of the findings.

9.      Lines 150-157: The authors superficially discuss the potential benefits of cannabinoids to treat pediatric cancers. However, it is unrealistic to discuss the potential use cannabinoids in pediatrics without mentioning their adverse effects on, for example, brain development. Please note that although some cannabinoid-based products (e.g., Dronabinol, nabilone) have been approved in the US for specific applications (and when other treatments have failed), these are only recommended for adults, as their safety and efficacy have not been established for patients under 18 years old.

10.   Lines 160-163: Where is the evidence for such effects? Were these data obtained in vitro, in vivo or in clinical trials? Details on the experiments (e.g., in vitro/in vivo model, type of cannabinoids tested, exposure settings, analyzed parameters) should be provided. The authors should also take into account that in vitro/in vivo data may not be necessarily translated into a clinical effect. Hence the importance of providing details on the specific effects observed in each study.

11.   Lines 164-165: please note that the use of drugs such as Dronabinol or Nabilone is only approved for cases in which other therapeutics fail.

12.   Lines 177-202: These two long paragraphs seem out of the scope of the manuscript, as there is no connection to cannabinoids.

13.   Lines 203-228: The authors only mentioned two clinical trials. However, Whiting et al. (JAMA 2015 313(24),2456-2473) have previously analyzed 28 clinical trials assessing the efficacy of cannabinoids for treating chemotherapy-related nausea and concluded that none showed any statistically significant effect of cannabi­noids compared to active comparators.

14.   Lines 237-246: Were such small differences statistically significant? It is not clear how such small differences suggest that marijuana and dronabinol could be useful in combination with other analgesics. Most importantly, from the authors’ description, it seems that non-users of marijuana were considered for testing, as all the 30 volunteers were daily marijuana smokers). As such, it is not possible to take any conclusions regarding the analgesic effect of marijuana. 

15.   Section 6: once again, the authors fail to provide details on the experiments (e.g., cannabinoids tested, concentrations, exposure settings) that would be important to support their conclusions. For example, it should be noted that most of the observed effects were only noted at high micromolar range concentrations, which may also be toxic to other cells.